# Variable opportunities for outcrossing result in hotspots of novel genetic variation in a pathogen metapopulation

Anna-Liisa Laine[1,2]*, Benoit Barrès[1†], Elina Numminen[1], Jukka P Siren[1,3]

[1]Research Centre for Ecological Change, Organismal and Evolutionary Biology, University of Helsinki, Helsinki, Finland; [2]Department of Evolutionary Biology and Environmental Studies, University of Zurich, Winterthurerstrasse, Switzerland; [3]Helsinki Institute for Information Technology, Department of Computer Science, Aalto University, Espoo, Finland

**Abstract** Many pathogens possess the capacity for sex through outcrossing, despite being able to reproduce also asexually and/or via selfing. Given that sex is assumed to come at a cost, these mixed reproductive strategies typical of pathogens have remained puzzling. While the ecological and evolutionary benefits of outcrossing are theoretically well-supported, support for such benefits in pathogen populations are still scarce. Here, we analyze the epidemiology and genetic structure of natural populations of an obligate fungal pathogen, *Podosphaera plantaginis*. We find that the opportunities for outcrossing vary spatially. Populations supporting high levels of coinfection –a prerequisite of sex – result in hotspots of novel genetic diversity. Pathogen populations supporting coinfection also have a higher probability of surviving winter. Jointly our results show that outcrossing has direct epidemiological consequences as well as a major impact on pathogen population genetic diversity, thereby providing evidence of ecological and evolutionary benefits of outcrossing in pathogens.
DOI: https://doi.org/10.7554/eLife.47091.001

*For correspondence:
anna-liisa.laine@ieu.uzh.ch

Present address: †Université de Lyon, Anses, INRA, USC CASPER, Lyon, France

Competing interests: The authors declare that no competing interests exist.

## Introduction

Many pathogens possess the capacity for sex – here defined in its broadest sense as the coming together of genes from different individuals (*Lehtonen and Kokko, 2014*) – despite being able to reproduce also asexually and/or via selfing. Individuals that undergo sexual reproduction transmit only half their genome per offspring produced in contrast to asexual and selfing individuals (*Lehtonen et al., 2012*) and hence, understanding the maintenance of sex is one of the fundamental challenges in evolutionary biology. To counteract this two-fold cost of sex, sexual outcrossing is assumed to provide both ecological and evolutionary advantages (*Otto, 2009*). The Red Queen Hypothesis predicts sexual reproduction to be advantageous in the presence of coevolving parasites, as offspring that are genetically different from their parents should have higher fitness than non-sexual offspring (*Bell, 1982*; *Hamilton, 1980*; *Lively, 2010*). In support of this prediction, empirical studies have demonstrated parasite mediated selection to explain the observed distribution of outcrossing in hosts (*King et al., 2011*; *Wilson and Sherman, 2013*; *Lively, 1987*). Just as sexual reproduction is expected to be selected for in hosts to evade parasitism, parasites should equally be under selection to generate novel genetic variation to infect their ever-changing host populations. Indeed, theoretically it has been possible to identify a parameter space where coevolution with the host favors sexual reproduction in the parasite (*Howard and Lively, 2002*; *Galvani et al., 2003*; *Salathé et al., 2008*). However, the empirical evidence for such advantages of sex in parasite populations are still few and conflicting (*Gouyon and de Vienne, 2015*).

**eLife digest** The existence of sex – broadly defined as the coming together of genes from different individuals – is one of the big evolutionary puzzles. Reproduction allows an organism to pass on its genes to future generations. However, while asexual and self-fertilizing individuals transmit all of their genes to their offspring, individuals that reproduce through sex transmit only half of their genome. This is considered the cost of sex.

Many pathogens reproduce through sex, despite often also being able to reproduce asexually or by self-fertilization. Typically a pre-requisite of sex in pathogens is for at least two different strains to infect the same host. Aside from this limitation, little is known about when, where and why pathogens have sex. It has been tricky to study due to the microscopic size of pathogens and the difficulties of identifying different sexes. Moreover, sexual reproduction may be triggered by environmental cues that are difficult to mimic under controlled experimental conditions.

Are there any benefits associated with pathogen sex? To find out, Laine et al. analyzed data collected over the course of four years from thousands of populations of a powdery mildew fungus that infected plants across the Åland islands. This revealed that the opportunities for pathogen sex vary in different locations. Areas where multiple strains of the fungus commonly infect the same plants result in hotspots of new genetic diversity. These mixed populations are also more likely to survive winter. This demonstrates the potential for pathogen sexual reproduction to provide an ecological benefit.

Identifying areas and populations where pathogens have sex can help to identify when and where new strains are most likely to emerge. In the future, studies that use similar methods to Laine et al. could help to predict where infections and diseases are highly likely to arise.

DOI: https://doi.org/10.7554/eLife.47091.002

Sexual reproduction may also confer ecological advantages by bridging unfavorable seasons or habitats. In free-living facultatively sexual organisms that alternate between asexual and sexual reproduction, the sexual offspring are often the dormant or dispersing life stages (*Stelzer and Lehtonen, 2016*; *Simon et al., 2002*). Similarly, in some of the most devastating fungal pathogens of crops, the spores that are produced via outcrossing are also those that are suitable for long-distance dispersal (*Rieux et al., 2014*), or provide means of surviving unfavorable environmental conditions (*Billiard et al., 2012*; *Burt, 2000*; *Saleh et al., 2012*). While sexual offspring do not contribute to current local population growth, the ability to outcross may be a key determinant of both the numeric and genetic composition of the next season's epidemic (*Penczykowski et al., 2015*; *Tack and Laine, 2014*). For such strategies where outscrossing is timed with low potential asexual growth, for example due to seasonality, the cost of sex is expected to be reduced (*Gerber et al., 2018*). Homothallic species, where a haploid individual may mate with other haploid individuals of its species, as well as with itself, may be considered a special case of facultative outcrossing. Here, the maintenance of outcrossing is particularly puzzling given that the offspring produced via haploid selfing are expected to yield the same ecological functions as those produced by outcrossing. Nonetheless, there is evidence of high rates of outcrossing in homothallic species (*Billiard et al., 2012*). To date the relevance of these short-term ecological processes in favoring selfing vs. outcrossing in pathogens populations has remained largely unknown.

Even when outcrossing is expected to provide short- or long-term advantages, maintenance of selfing may be favored when mate availability is spatially and/or temporally variable (*Jarne and Charlesworth, 1993*). For many pathogens, coinfection is the ecological prerequisite of sexual recombination, as outcrossing and hybridization take place during active infection of the same host individual by different strains (*Froissart et al., 2005*). With molecular tools becoming increasingly available for pathogens, we are beginning to unravel the spatio-temporal distribution of coinfection as well as the ecological outcomes, which may range from facilitation to competition (*Tollenaere et al., 2016*). Although coinfections have been widely reported for different pathogens, remarkably little is understood of the determinants of coinfection (*Tollenaere et al., 2016*). Identifying factors that increase the probability of coinfection also shed light on where we expect to see outcrossing in pathogens with mixed mating strategies.

Mixed reproductive strategies have been described for a wide range of pathogen species (*Billiard et al., 2012*; *Billiard et al., 2011*). However, the sexual stage is methodologically notoriously difficult to study in many pathogens given their microscopic size and the fact that mating types cannot be identified morphologically. Moreover, sexual reproduction may take place inside the host, and may be triggered by specific environmental cues that are difficult to mimic under controlled experimental conditions (*Billiard et al., 2012*; *Tack and Laine, 2014*). Hence, remarkably little is understood of this critical life-history stage. To understand why outcrossing is broadly maintained in pathogens despite the costs, here we investigate the ecological and genetic consequences of putative outcrossing in a large natural pathogen metapopulation. Our analysis is based on data collected from *Podosphaera plantaginis*, a specialist powdery mildew fungus naturally infecting *Plantago lanceolata*. The visually conspicuous symptoms caused by *P. plantaginis* enable accurate tracking of infection in the wild. Long-term epidemiological data across approximately 4000 local plant populations in the Åland Islands, southwest of Finland, have demonstrated this pathogen to persist as a highly dynamic metapopulation with frequent extinctions and (re)colonizations of local populations (*Jousimo et al., 2014*). Overwinter survival of local pathogen populations has proven to be the vulnerable life-history stage of *P. plantaginis* in Åland with a high fraction of the local pathogen populations going extinct (*Jousimo et al., 2014*). The pathogen survives the winter in resting structures, chasmothecia (*Tack and Laine, 2014*). These resting structures are produced through sexual reproduction as the hyphal cells of one (selfing) or two strains (outcrossing) fuse when infecting the same host plant. The resulting diploid zygote undergoes meiotic division to yield haploid ascospores that develop inside the chasmothecium. At the onset of the growing season these chasmothecia rupture, releasing the ascopores that initiate new infections (*Tollenaere and Laine, 2013*). Here, we (i) determine how coinfection - the pre-requisite for sexual outcrossing - varies in natural pathogen populations. We then measure whether (ii) putative outcrossing (*i.e.* coinfection) is associated with the generation of novel pathogen multilocus genotypes, and (iii) increased pathogen population overwintering success (*Jousimo et al., 2014*). We've surveyed and sampled all found pathogen populations in the Åland Islands for four consecutive years, and we use Spatial Bayesian models (Integrated Nested Laplace Approximation; INLA; *Lindgren and Rue, 2015*) to analyse data on disease dynamics and genotypic diversity from the natural metapopulation.

## Results

We first quantify how the opportunities for sexual outcrossing – thta is coinfection - vary across hundreds of wild pathogen populations in four consecutive seasons. We sampled 619, 703, 693 and 833 populations in 2012–15 for subsequent genotyping (Table S1). We used a SNP genotyping protocol to estimate the number of multilocus genotypes (MLGs) and prevalence of coinfection within pathogen populations (*Tollenaere et al., 2012*). Coinfection proved to be common yet spatially variable across the *P. plantaginis* metapopulation (*Figure 1A*) (*Susi et al., 2015*). In all years approximately half of the pathogen populations supported at least one coinfected sample, (45–58%; *Supplementary file 1*). We found that coinfection was more likely to be found in larger and more diverse pathogen populations (Significant positive effect of number of MLGs and infection abundance; *Table 1*). Connectivity of pathogen populations, which is considered a proxy for gene flow among populations as it is estimated from distances separating local pathogen populations (*Jousimo et al., 2014*), had a positive, albeit not significant, effect on the probability of coinfection (*Table 1*). The INLA model we use here, controls for spatio-temporal autocorrelation characteristic of spatial ecological data due to unmeasured variables, thereby providing a conservative estimate of the model parameters as evidenced by model validation checks (*Figure 1—figure supplements 1–5*) (*Lindgren and Rue, 2015*).

We then used an Approximate Bayesian Computation (ABC) approach to determine whether we detect more coinfection within populations than would be expected based on the number of parasite genotypes and host availability. Our results show that an already infected plant is more likely to be infected by another strain (*Figure 1—figure supplement 6*). In other words, coinfections were more common than expected by chance under the assumption that infections by different MLGs are statistically independent. The result was consistent in both years 2012 and 2013 (posterior probability 0.98 and 0.988, respectively; *Figure 1—figure supplement 6*), with the parameter controlling the

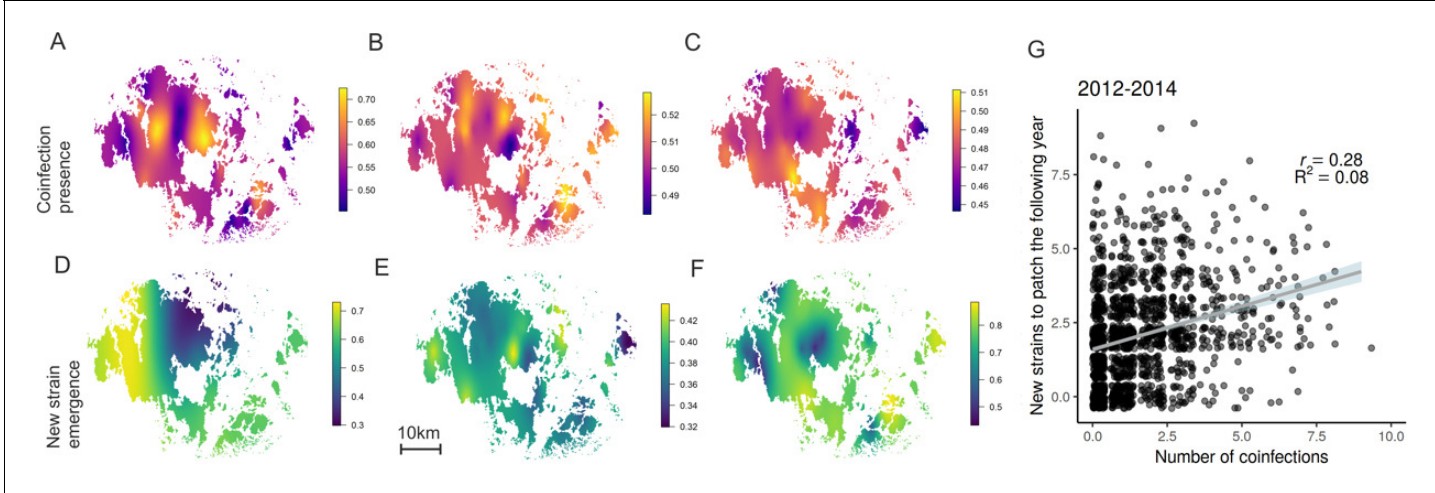

**Figure 1.** Hotspots for pathogen outcrossing - Novel pathogen genotypes emerge in populations with high prevalence of coinfection the previous year. The heatplots for the spatial occurrence probability of coinfection in 2012–14 (A-C, respectively), and probability of a novel MLG emerging in pathogen populations in 2013–15 (D-F, respectively), were obtained by fitting spatial intercept-only models as presented in *Table 1*. The probabilities of events for each location were obtained from the linear predictor based on the estimated intercept and the spatial random field. Warmer tones correspond to areas of higher event occurrence probabilities. The number of genotyped samples supporting coinfection in a pathogen population correlated positively with the probability of detecting a MLG that had not been previously detected in that population using all data collected in 2012–15 (G). The fitted line corresponds to the 95% confidence level interval generated by the linear model in *Table 1*. Darker shades indicate higher data density.

DOI: https://doi.org/10.7554/eLife.47091.003

The following figure supplements are available for figure 1:

**Figure supplement 1.** The estimated posterior distributions of the spatial ranges for the fitted spatio-temporal models described in *Table 1*.

DOI: https://doi.org/10.7554/eLife.47091.004

**Figure supplement 2.** Mesh used for the spatio-temporal model.

DOI: https://doi.org/10.7554/eLife.47091.005

**Figure supplement 3.** PIT-value distributions.

DOI: https://doi.org/10.7554/eLife.47091.006

**Figure supplement 4.** CPO-values distributed in space.

DOI: https://doi.org/10.7554/eLife.47091.007

**Figure supplement 5.** Distributions of the CPO-values for the four models for the different years.

DOI: https://doi.org/10.7554/eLife.47091.008

**Figure supplement 6.** Posterior distributions of the parameter values from the ABC inference on the infections and co-infections.

DOI: https://doi.org/10.7554/eLife.47091.009

**Figure supplement 7.** Accuracy of the parameter estimates from the ABC analysis of infections and co-infections.

DOI: https://doi.org/10.7554/eLife.47091.010

**Figure supplement 8.** Posterior distributions of the parameter values from the ABC inference on the infections and co-infections under different modeling assumptions using data for 2012.

DOI: https://doi.org/10.7554/eLife.47091.011

**Figure supplement 9.** The number of samples collected from *Podosphaera plantaginis* populations in 2013–15 depicted according to population size categories.

DOI: https://doi.org/10.7554/eLife.47091.012

**Figure supplement 10.** The global allele frequencies of the 19 SNP loci used for genotyping.

DOI: https://doi.org/10.7554/eLife.47091.013

**Figure supplement 11.** A schematic representation of the algorithm that identifies the unique multilocus genotypes that form the observed coinfections.

DOI: https://doi.org/10.7554/eLife.47091.014

prevalence of coinfections, $\gamma$, being reliably estimated under different modeling assumptions (*Figure 1—figure supplement 7–8*).

The detection of novel MLGs in the pathogen metapopulation from one year to another suggests that sexual outcrossing is common for this pathogen. When all located pathogen populations were

**Table 1.** The estimated posterior means and 95% credibility intervals for the parameters of the fitted spatio-temporal models. In each model, the abundance of infection category one is considered as the baseline factor level for every model. Whenever the credibility interval does not include zero (denoted with bold), the effect is considered significant. For temporal and spatial range, and the nominal variance, significance cannot be estimated as they can only get positive values. For more information about the predictors, please see the Model Variables-section in the Methods.

| | Model | | |
|---|---|---|---|
| Parameter | Coinfection presence in a pathogen population (0/1) | Number of new strains within a population | Successful pathogen population overwintering (0/1) |
| Intercept | −1.04, (−1.16,−0.92) | −0.38, (−0.49,−0.27) | 1.17, (0.93, 1.41) |
| Number of coinfections | *not fitted* | 0.06, (0.02, 0.11) | 0.28, (0.11, 0.46) |
| Number of strains | **1.07, (0.93, 1.21)** | **0.08, (0.04, 0.13)** | **0.34, (0.18, 0.5)** |
| Abundance of infection (category 2) | **0.37, (0.22, 0.53)** | −0.03, (−0.08, 0.02) | **0.39, (0.27, 0.51)** |
| Abundance of infection (category 3) | **0.61, (0.46, 0.75)** | −0.04, (−0.1, 0.01) | **0.55, (0.39, 0.72)** |
| Year 2013 | *not fitted* | **0.17, (0.03, 0.32)** | 0.05, (−0.27, 0.37) |
| Year 2014 | *not fitted* | **1.45, (1.33, 1.58)** | **0.52, (0.19, 0.85)** |
| Pathogen connectivity ($S_i^p$) | 0.05, (−0.08, 0.19) | 0.02, (−0.02, 0.07) | −0.04, (−0.19, 0.1) |
| Host population size (log$m^2$) | 0.01, (−0.12, 0.14) | **0.09, (0.04, 0.14)** | 0.03, (−0.1, 0.16) |
| Temporal autocorrelation ($\varphi$) | 0.03, (−0.41, 0.43) | 0, (−0.41, 0.41) | −0.08, (−0.43, 0.28) |
| Spatial range (meters) | 11410, (460, 50811) | 14229, (3266, 38058) | 8512, (1775, 25860) |
| $\sigma_2$ (Nominal variance) | 0.2, (0, 1.33) | 0.07, (0.01, 0.22) | 0.45, (0.1, 1.28) |

DOI: https://doi.org/10.7554/eLife.47091.015

genotyped, we identified 182, 189, and 235 novel MLGs in 2013–2015, respectively, when compared to MLGs detected the previous year (*Supplementary file 1*; *Figure 1B*). The number of new MLGs at the population level increased with the prevalence of coinfection at the end of the previous epidemic season which is the time when outcrossing takes place (*Figure 1C*; *Table 1*). The prevalence of coinfection was a strong predictor of novel MLGs the following year even after controlling for the effects of local pathogen population size, diversity (number of MLGs), pathogen population connectivity (proxy for gene flow) as well as spatial and temporal autocorrelation (*Table 1*).

Using data from the natural pathogen metapopulation, we also found that in those pathogen populations where the prevalence of coinfection is high – and hence where sexual reproduction can take place – the pathogen has a higher survival probability (*Table 1*; *Figure 2A and B*). The effect of coinfection on successful overwintering is positive even after controlling for the effects of pathogen population size and diversity, which both increase survival probability. The INLA model also controls for spatial and temporal autocorrelation in these data that may be generated by abiotic variation known to be important for overwintering ecology of this pathogen (*Penczykowski et al., 2015*). Hence, we view this as a conservative estimate of the effect of coinfection on overwintering. Resting spores, which were visually scored in the field (*Tack and Laine, 2014*), were produced in nearly all pathogen populations regardless of whether they supported coinfection or not (96% *vs.* 93%, respectively).

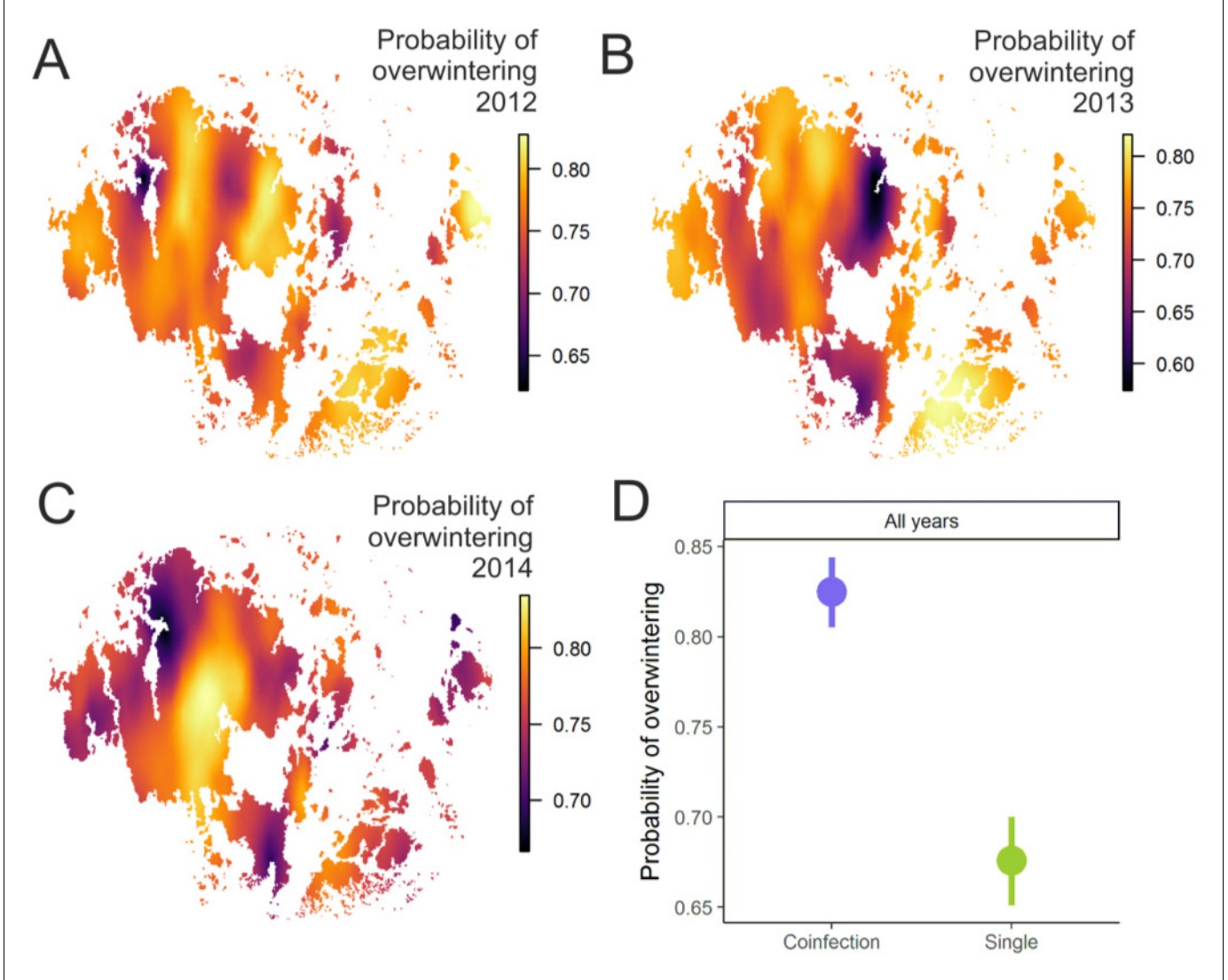

**Figure 2.** Pathogen overwinter survival was highest in populations that supported higher levels of coinfection. The heatplot for the spatial occurrence probabilities of overwinter survival were obtained by merging the corresponding event data from corresponding years and fitting spatial intercept-only models as presented in *Table 1*. The probabilities of events for each location were obtained from the linear predictor based on the estimated intercept and the spatial random field. The maps shows spatial variation in overwinter survival for 2012–13 (A), 2013–14 (B) and 2014–15 (C) with warmer tones corresponding to areas of higher event occurrence probabilities. (B) The average probability of successful pathogen population overwintering was higher in populations with coinfection than in populations without coinfection in all three years in 2012–15.

DOI: https://doi.org/10.7554/eLife.47091.016

## Discussion

Here, we report compelling evidence of outcrossing that generates novel genetic diversity in the pathogen metapopulation. Our results demonstrate variation in how opportunities for outcrossing are distributed across space. Finding more coinfection than would be expected by chance is in line with previous fine-scale field sampling of infections and experimental work, which show that hosts already infected with one strain of the pathogen are more likely to become infected by another strain of the same pathogen than uninfected hosts (*Laine, 2011*; *Susi and Laine, 2017*). This may be due to already infected individuals becoming more susceptible to subsequent infection, or due to strains aggregating on those hosts that are the most susceptible genotypes (*Susi and Laine, 2017*). Moreover, variation in host density and (micro)climatic conditions may be an important driver of infection patters in the wild (*Penczykowski et al., 2018*). Our results do not support the priming

hypothesis (*Hilker et al., 2016*), whereby prior attack provides increased protection against later attack.

We find that spatial variation in coinfection results in spatially delineated hotspots of novel genetic diversity. The high number of new MLGs detected every year is indicative of outcrossing taking place in this pathogen metapopulation. Although our sampling is likely to miss some rare strains, and novel MLGs may be generated through mutations, these are unlikely to explain the high turnover of MLGs between years we report here. Mainland populations, which are separated by at least 38 km of open water, are also expected to play a negligible role as sources of gene flow given that experimental and field data have confirmed this pathogen to typically disperse short distances (*Jousimo et al., 2014*; *Tack et al., 2014*). Our results suggest that sexual outcrossing takes place where there is the opportunity for it, that is in populations where levels of coinfection are high. To date, this phenomenon has only received limited experimentally derived support in pathogens (*Schelkle et al., 2012*). Spatio-temporal variation in outcrossing is expected to have both evolutionary and epidemiological consequences for the pathogen. In the short term, generation of novel genetic diversity may increase transmission across host populations that support considerable resistance diversity both within and among populations (*Jousimo et al., 2014*; *Laine et al., 2011*). Novel genetic diversity may also increase the evolutionary potential of pathogens that need to adapt to both biotic and abiotic variation in their environment (*Greischar and Koskella, 2007*; *Wolinska and King, 2009*).

Our multi-year census data further revealed the putative outcrossing to yield a benefit that is realized in an ecologically important function – higher overwintering success. Overwintering determines both the genetic and numeric structure of the next epidemic (*Tack and Laine, 2014*; *Penczykowski et al., 2015*), and hence may be a sufficiently important trait to promote the maintenance of outcrossing in a pathogen that is able to complete its life-cycle also through haploid selfing. Our results suggest that successful overwintering is not due to higher production of resting spores. Hence, there may be a difference in the quality of progeny produced via selfing *vs.* outcrossing. Prior experimental work has demonstrated significant variation in spore viability in the resting structures of *P. plantaginis*. There is evidence of higher viability of progeny from coinfections than from single infections, but the strength and direction of this trend is affected by the genotypes of the interacting strains, as well as by temperature (*Vaumourin and Laine, 2018*). Despite the higher overwintering success of outcrossed progeny, haploid selfing may be preserved due to the low probability of encountering a suitable mating partner infecting the same host. Moreover, there may be a cost to outcrossing as it breaks up locally adapted pathogen populations by producing novel – and potentially maladapted – genetic variation.

Overall, the selection pressures and opportunities to mate vary considerably across space and time, and hence, it is not surprising that many pathogens have evolved highly complex mating strategies (*Billiard et al., 2011*). A loss of sexual reproduction in pathogens has been linked to homogenous habitat (*Saleh et al., 2012*) or stable environmental conditions (*Barrett et al., 2008*). Maintaining a mixed mating system may provide a bet-hedging strategy for this pathogen to survive in a fragmented landscape, with a high probability of population extinction during the off-season. It is noteworthy that here we succeeded in identifying predictors of how coinfection is spatially distributed - and hence where hotspots of outcrossing are formed - despite the considerable environmental 'noise' this natural system supports. The correlations in field collected data we have observed here are a promising start to uncovering the variable selective pressures and advantages of outcrossing in pathogens. Establishing direct links between variation in reproductive strategies and epidemiological dynamics offers an exciting venue of research, and is needed to truly predict where risks of infection and disease emergence are the highest.

## Materials and methods

### Survey of natural pathogen metapopulation

*Plantago lanceolata* is a perennial rosette-forming herb that is naturally infected by *Podosphaera plantaginis* (Castagne; U. Braun and S. Takamatsu), a powdery mildew fungus in the order Erysiphales within the Ascomycota. This pathogen is a host-specific obligate biotroph that completes its entire life cycle on the surface of the host plant where it is visible as localized (nonsystemic) white

powdery lesions. The interaction between *P. lanceolata* and *P. plantaginis* functions in a two-step manner typical of many plant–pathogen associations. First, as the pathogen attempts to infect a new host, the interaction is strain specific as a given host genotype expresses resistance against some strains (recognition) of the pathogen while being susceptible to others (nonrecognition) (*Jones and Dangl, 2006*). Once a *P. plantaginis* strain has successfully established there is still considerable variation in its development that is affected by both pathogen and host genotype (*Laine, 2007*). The pathogen is a significant stress factor for its host and may cause host mortality (*Penczykowski et al., 2015*).

The locations of *P. lanceolata* populations have been systematically mapped in the Åland Islands, southwest of Finland, since the 1990s. There are currently *c.* 4000 known host populations that range in size from a few square meters to several hectares, with a median size of 300 m$^2$ (*Jousimo et al., 2014*). Within host populations, initial pathogen foci are established from resting spores (chasmothecia), or from a spore immigrating into the local population from another population. The first visible signs of infection appear in late June as white-greyish lesions consisting of mycelium supporting spores (conidia) are formed. The spores are dispersed by wind to the same or new host individuals. Some six to eight clonally produced generations (estimated from spore germination-production times observed in the laboratory) follow one another in quick succession, often leading to a substantial proportion of the host individuals within a population being infected by late summer (*Ovaskainen and Laine, 2006*). Resting spores (chasmothecia) appear towards the end of the growing season in August–September. Each chasmothecia contains eight ascospores that can each cause a new infection in the spring upon their release. Infected leaves may support hundreds of chasmothecia. In *P. plantaginis*, chasmothecia production is achieved via both haploid selfing as well as outcrossing between two strains simultaneously infecting the same host plant. Pure strains of *P. plantaginis* have been shown to carry both MAT1-1-1 and MAT1-2-1 that determine compatibility in several other powdery mildew species (*Tollenaere and Laine, 2013*).

In early September every year since 2001, all known *P. lanceolata* populations have been surveyed for the presence/absence of the powdery mildew (for details on the survey, please see *Jousimo et al., 2014*). These data can be used to identify pathogen populations, which have persisted from one year to the next, newly colonized populations, and populations that have gone extinct. These data have demonstrated that *P. plantaginis* persists as a highly dynamic metapopulation through extinction and (re)colonization of local host populations (*Jousimo et al., 2014*).

## Genotyping of field collected pathogen samples

In 2012–15 nearly all located pathogen populations were sampled for genotyping (N = 619, 703, 693, and 833 populations, respectively, which represented 96–97% of all located pathogen populations each year; *Supplementary file 1*). A sample consists of one infected leaf collected from an infected plant, and infected plants were sampled at a minimum distance of five meters between infected plants. The aim was to collect ten samples from each population but in smaller pathogen populations sampling effort needed to be scaled to how much infection was available for sampling (Please see 'Pathogen population size' below in Model variables -section below for a description, and for numbers of samples in each pathogen population size category, please see *Figure 1—figure supplement 9*). The infected leaves were placed in separate falcon tubes and brought back to the laboratory where fungal material for each sample was collected by scraping off the surface of the infected leaf. This material and a 1 cm$^2$ piece of the same infected leaf were placed in an individual well of a 96-well plate. Samples were stored at −20°C until DNA extraction.

DNA extraction was performed using E.Z.N.A. Plant DNA kit (Omega Bio Tek Inc, Norcross, GA, USA) at The Institute of Biotechnology (BI, Helsinki, Finland). Samples were genotyped with 27 SNP markers using Sequenom MassARRAY iPLEX platform as described in *Tollenaere et al. (2012)* at the Finnish Institute for Molecular Medicine (FIMM, Helsinki, Finland). Automatic calling of the genotypes was performed using MassARRAY Typer four software (Sequenom, San Diego, CA). Because of the presence of null alleles in the studied populations, eight SNP were discarded from the analysis. Allele frequencies are shown in *Figure 1—figure supplement 10*. The genotyping results were used to identify the multilocus genotypes (MLGs) of each sample and to detect coinfection in the collected samples. *Podosphaera plantaginis* is haploid, and therefore the detection of a heterozygote genotype for one or more SNP markers is a clear highly repeatable method for calling coinfections (*Tollenaere et al., 2012*; *Susi et al., 2015*).

## Model variables

The survey data from the natural populations and the genotyping data from years 2012–15 were used to generate the following variables used in the analyses (See Statistical modeling below):

### Pathogen population size

The abundance of pathogen was scored on a categorical scale with five levels, defined as: 0 = no infection, 1 = 1–9 infected plants, 2 = 10–99 infected plants, 3 = 100–999 and 4 = 1000 or more. Due to the small number of category four infections, the categories 3 and 4 were merged for the analyses.

### Number of MultiLocus genotypes (MLG) in local pathogen populations and identification of MLGs from coinfected samples

For each population and year we estimated the number of unique multilocus genotypes (MLG) observed using the SNP genotyping protocol described above. This number was obscured by samples supporting coinfection, for which the MLGs could not be directly identified. To solve this, we developed an algorithm that gives as an output a lower limit for the number of strains in the population, that would correspond to all the observed MLG profiles, both those observed for single and coinfected samples (*Figure 1—figure supplement 11*). The algorithm was designed to be conservative to avoid overestimating the number of MLGs. To this end, we assumed that in a coinfected sample, the detection of multiple alleles in a locus could have failed, and only one allele observed. The five stages of the algorithm are as follows: First, we identified the unique MLGs from the single infections. Second, we identified coinfections, whose alleles could not be obtained as a combination of two unique MLGs from the first stage. Third, we removed duplicate coinfections, whose alleles were subsets of another coinfection in the loci that the samples shared. We considered subsets to accommodate possible genotyping errors for the coinfections, where only one allele is detected for a loci. Fourth, for each remaining coinfection, we constructed a new minimal candidate MLG, which was needed in addition to a unique MLG identified from single infections to generate the alleles in the coinfection. Only those loci where an allele not present in the single infection was observed in the coinfection were defined for the candidate MLG, and the remaining alleles were defined as unknown, because they could have been unobserved due to genotyping error. Fifth, we merged candidate MLGs that had same alleles in all the loci that were defined. The unique MLGs from single infections and the constructed candidate MLGs were now considered the unique strains for the population. The number of new MLGs in a population for a given year was estimated using the same algorithm. First, we applied the algorithm to samples from current and previous year, and then to samples from current year only. The number of new MLGs was then estimated as the difference between the number of MLGs from the two runs.

### Prevalence of coinfection in a population

Number of samples supporting coinfection in local pathogen populations identified using the SNP genotyping protocol described above.

### Abundance of pathogen resting structures

The powdery mildew survives the winter in resting structures (chasmothecia). These are visually conspicuous and their abundance has been systematically surveyed on the infected plants located during the field survey (*Tack and Laine, 2014*). The abundance of chasmothecia in local populations has been recorded on a categorical scale: 0 = no infected plants with chasmothecia; 1 =<10% of infected plants with chasmothecia; 2 = 10–25% of infected plants with chasmothecia; 3 = 25–50% of infected plants with chasmothecia, and 4 = 50–100% of infected plants with chasmothecia.

### Successful overwintering of pathogen populations

Pathogen populations that were located in two successive autumn surveys were considered to have successfully overwintered.

## Pathogen connectivity

Connectivity, denoted with $S_i^p$ [37] of each pathogen population, $i$, was computed as:

$$S_i^p(t) = \sum_{i \neq j} e^{-\frac{d_{i,j}}{\alpha}} O_j$$

where $O_j = 1$, if population $j$ was infected and $O_j = 0$, if population j was not infected in the current year t, and $d_{i,j}$ corresponds to the distance between populations i and j. The parameter $\alpha$, describing the mean dispersal distance at the metapopulation level, was set $\alpha = 1000$ (meters) following results of *Jousimo et al. (2014)*. *Si* is thus assumed to be a rough proxy for the expected number of pathogen transmission coming to population *i* from all the infected populations, under the assumption of exponentially distributed dispersal distance.

## Host population size

The total coverage (in m$^2$) of the host in the local population has been visually assessed each year simultaneously when the pathogen infection data were collected. The criteria for describing host populations is provided in detail in *Ojanen et al. (2013)*. In brief, suitable host populations have been identified by the presence of dense clusters of *P. lanceolata*, which in Åland occur on dry meadows and pastures. These occur as well-defined, discrete patches across the landscape.

## Spatial bayesian modelling of the effect of coinfection on pathogen overwintering and strain diversity

We modeled the following events of interest: the effect of the number of coinfections on persistence of infection from one year to the next, and the number of new, previously unidentified, MLGs in a pathogen population that survived from one year to the next. In addition, we assessed the drivers of the presence/absence of resting spores and coinfections among the infected populations. Our models were fitted to data from years 2012-2015. To control for the possible effect that population size, connectivity and diversity could have on the results, our models included the following predictors (described above in detail): Pathogen population size, host population size, number of distinct pathogen strains, and pathogen connectivity $S_i^p$. Predictors with continuous support and the number of observed coinfections, were scaled and centered around zero, and factors transformed into binary 0/1-variables.

### Spatio-temporal logistic regression model

The statistical modeling of the phenomena of interest (coinfection presence, the emergence of new strains in a population and survival of pathogen population) was done by considering a logistic regression modeling framework, where the observation at location *s* at time *t* was assumed to be Bernoulli distributed with the event probability $\mu_{\mathrm{st}}$. The spatial- and temporal autocorrelation was taken into account by assuming that $\mu_{\mathrm{st}}$ has an explicit spatio-temporal correlation structure, defined as follows:

$$\mathrm{logit}(\mu_{st}) = z(s,t)\beta + \delta(s,t) + \varepsilon(s,t)$$

$$\delta(s,t) = \varphi \cdot \delta(s,t-1) + w(s,t)$$

Here z(s,t) corresponds to the covariate information and $\varepsilon(s,t)$ is the measurement error. Further, $\delta(s,t)$ is a spatio-temporal latent process with first-order temporal autocorrelation, described by parameter $\varphi(|\varphi| <; 1)$, and spatially correlated outcomes described by zero-mean Gaussian distribution $w(s,t)$. The spatial correlation is included by assuming that $w(s,t)$ has the following covariance structure:

$$\mathrm{Cov}\left(w(s,t), \mathrm{Cov}\left(s^{'},t^{'}\right)\right) = \begin{cases} \sigma_2 C(h) \\ 0, \ if \ t \neq t' \end{cases}$$

Here $\sigma_2$ is the overall variance of the random field, and $C(h)$ is the Matern covariance function, that only depends on the Euclidean distance *h* between the latent locations *s* and *s'*, and hyperparameters $\kappa$ and $\vartheta$. Details on the covariance function, and an in-depth description of the modeling

framework is given in *Cameletti et al. (2013)*. The aim of our analysis was to infer the joint posterior distribution of $\theta$:

$$\theta = \{\beta, \varphi, \sigma_2, \kappa, \vartheta,$$

the main interest being in $\beta$, which describes the effects of the covariates listed and described in the Model variables section.

Efficient implementation of such inference is provided by the R-INLA package, which provides posterior distribution of $\theta$ by marginalizing $\mu_{st}$ with Laplace approximation. When setting the prior distributions for hyperparametersand $\vartheta$, we followed the heuristics proposed by the INLA developers (*Lindgren and Rue, 2015*), ensuring that a 95% prior probability is set to the range being smaller than the size of the spatial domain considered. Here the range is defined as the distance at which the spatial autocorrelation becomes negligible (smaller than 0.1 under the Matern covariance). The other prior distributions were set to default uninformative prior distributions (*Lindgren, 2012*), Finally, another computational gain is obtained by approximating the spatial random field $w(s,t)$ with the help of a mesh, visualized in *Figure 1—figure supplement 1*.

## Model validation

As a model validation argument, we inspected the distribution of the CPO-values, *Conditional Predictive Ordinates*, defined as:

$$CPO_i = P\left(d_i^{new} = d_i \mid D_{-i}\right).$$

$CPO_i$ 'corresponds to the probability of predicting the observation $d_i$, when $d_i$ is excluded from the model fitting. Briefly, high CPO-values indicate high predictive power of the fitted model, while very low CPO values would indicate outlier observations, or a poor fit of the model, and patterns in CPO values (such as spatial or temporal) suggest inappropriate model structure.

To ensure the adequacy of the fitted models, we considered leave-one-out predictive measures of fit, assessing how well the model is able to predict the observed dynamics. In particular, we considered the so-called *probability integral transform* (*Dawid, 1984*), defined as:

$$PIT_i = P\left(d_i^{new} \leq d_i^{new} \mid D_{-i}\right),$$

Here $d_i$ denotes for the i'th observed outcome in the data, and $D_{-i}$ for the data with this observation excluded, and $PIT_i$ corresponds to the cumulative predictive distribution for the outcome i, given all the other observations. We adjust this to take into account the discrete nature of the modeled outcomes, as follows (*Czado et al., 2009*):

$$PIT_i = P\left(d_i^{new} <; d_i \mid D_{-i}\right) + \frac{1}{2} * P\left(d_i^{new} = d_i \mid D_{-i}\right).$$

In brief, skewedness of the distribution of PIT-values indicates biases, U- and inverse-U-shaped distributions indicate under- and over-dispersion, respectively, while a uniform distribution indicates good model fit.

## Parameter estimates

We considered a covariate to have a significant effect on the outcome, whenever its 95% credibility interval did not contain zero.

## Analysis of occurrence of co-infections with approximate bayesian computation

We take an Approximate Bayesian Computation approach to determine whether there is more co-infection in the pathogen metapopulation than expected by chance. The model does not take into account the spatial structure of the patches, but considers them independent conditional on the model parameters.

## Process model

First, consider a single patch $i$. Let $x_i$ denote a vector of environmental covariates of the patch $i$ and including a constant. The number of plant individuals in patch $i$, $N_i$, is modeled as

$$N_i \sim Poisson(x_i \alpha_N),$$

where $\alpha_N$ are the parameters relating covariates $x_i$ to the expectation of $N_i$. The number of MLGs in patch $i$, $M_i$, is modeled similarly

$$M_i \sim Poisson(x_i \alpha_M),$$

where $\alpha_M$ are the parameters. The number $M_i$ might be restricted with an upper limit $M_{max}$, so that $M_i \leq \sim M_{max}$.

Let $\mu_i = (\mu_{i,1}, \ldots, \mu_{i,M_i})$ be a vector of transformed prevalences of the different MLGs in patch $i$. We assume a normal distribution for $\mu_{i,k}$, $k=1,\ldots,M_i$,

$$\mu_{i,k} \sim Normal(x_i \beta, 1),$$

where $\beta$ is a vector of linear predictors.

For the $j$th plant in patch $i$ the infection status is represented by a vector $Z_{i,j} = (Z_{i,j,1}, \ldots, Z_{i,j,M_i})$. If $Z_{i,j,k} >; 0$, then the plant if infected by MLG $k$, and otherwise it is not infected by the strain $k$. We assume that the plant is exposed to the MLGs $1,\ldots,M_i$ in order $\sigma_{i,j}$, which is a permutation of the MLG indices. The distribution for $Z_{i,j,k}$ is

$$Z_{i,j,k} \sim Normal(\eta i,j,k, \tau^2),$$

where $\tau^2$ is the variance parameter and the mean $\eta_{i,j,k}$ depends on the values of $Z_{i,j,k}$ for MLGs l that are before MLG k in the order $\sigma_{i,j}$ . Specifically,

$$\eta_{i,j,k} = \begin{cases} \mu_{i,k} + \gamma & \text{if } Z i,j,\sigma i,j(l) >; 0 \text{ for any } \sigma i,j(l) \ lt;k \\ \mu_{i,k} & \text{otherwise} \end{cases}$$

In other words, if MLG $k$ infects the plant (i.e. $Z_{i,j,k} >; 0$), then the mean $\eta_{i,j,k}$ is increased by $\gamma$ for the MLG that the plant is exposed afterwards according to the order $\sigma_{i,j}$. If $\gamma >; 0$, then it is easier for the subsequent strains to infect the plant, while if $\gamma >; 0$, then it is more difficult for the subsequent strains to infect the plant.

The order $\sigma_{i,j}$ may be same for all plants in the patch, so that $\sigma_{i,j} = \sigma_i$ for all plants $j$ in patch $i$. The order $\sigma_{i,j}$ may be given a uniform distribution in the set of all permutations, or it may depend on the strain prevalences $\mu_i$.

## Sampling model

The data considered here are the genotypes of sampled individuals. For a single patch $i$, if the mildew is present in the patch, up to 10 infected plants are sampled and the MLGs infecting them determined. The exact number of sampled plants depends on the number available in the patch; as many infected plants are sampled as possible until 10 is reached. The observed genotype for the $j$ th sampled plant (note that here $j$ goes from 1 to 10, not from 1 to $N_i$) in patch $i$ is

$$G_{i,j} = \begin{cases} k \text{ if } Z_{i,j,k} > 0 \text{ and } Z_{i,j,l} < 0 \text{ for all } l \neq k \\ 0 \ Z_{i,j,l} > 0 \text{ for multiple } l \end{cases}$$

In words, genotype $G_{i,j}$ is the MLG index, if only one MLG is infecting the plant, and 0 if there is co-infection by multiple MLGs. Note that the genotypes of the MLGs in a co-infection are not observed.

We also use data on pathogen population size for patch $i$, $AA_i$, measured as a categorical variable (1 = less than 10 plants infected, 2 = 10-99 plants infected, 3 = 100-999 or 4 = 1000 or more) and infection prevalence for patch $i$, $RA_i$, (categories of the proportion of plants infected: 1 = 0-0.25, 2 = 0.25-0.5, 3 = 0.5-0.75 and 4 = 0.75-1).

## Prior distributions

The model includes $2 + 3n_x$ parameters, where $n_x$ is the number of covariates $x_i$ for patch $i$, including constant. All of the parameters, or alternatively their logarithms, are given uniform prior distributions on bounded intervals. For example, the variance parameter $\tau^2$ has to be positive, so the uniform distribution is put on its logarithm. Additionally, some covariates might be restricted to have a positive (negative) effect on some feature, in which case the uniform distribution is put on the logarithm of the parameter (negative of the parameter).

## Posterior distribution

The goal of the inference is to compute and sample from the posterior distribution of the parameters given the observed data,

$$p(\gamma, \tau^2, \alpha_N, \alpha_M, \beta | G, AA, RA)$$

As data is available for multiple years, we considered each of them separately. As covariates in $x_i$ for patch $i$ we included the area of the patch and *P. lanceolata* coverage. We included in the data only patches, where these covariates are available, resulting in 3817 and 3615 patches for years 2012 and 2013, respectively. The covariates (constant excluded) were standardized to have zero mean and unit variance across the patches. The total number of parameters in the model is 11.

Unless otherwise stated, the maximum number of MLGs in each patch was set to $M_{max} = 30$. Similarly, the order $\sigma_{i,j}$ was given a uniform distribution in the set of all permutations independently for each $i$ and $j$.

## Approximate bayesian computation

We use an Approximate Bayesian computation (ABC) approach (*Beaumont, 2010*) to make inference on the posterior distribution of the parameters. ABC refers to a class of computational methods for Bayesian statistics, where the computation of likelihood is replaced by simulations from the model, and it is ideal for our purpose given that the dimension of the model is not fixed, because the number of plants $N_i$ and the number of MLGs $M_i$ in patch $i$ is allowed to vary. Moreover, as our aim is to infer general patterns from the data in using information on for example the number of MLGs in a particular patch, an ABC approach is an efficient strategy.

Parameter estimation is carried out using standard ABC techniques (*Beaumont, 2010*). Candidate parameters are simulated from the prior, pseudo-datasets simulated for each parameter value and summary statistics are computed from the datasets. The number of simulations in each analysis is 100,000. The threshold of accepting simulations is based on obtaining a fixed size set of accepted values, that is the acceptance threshold is a quantile of the distances. We use 500 closest values in the analysis.

Raw summary statistics are transformed using two transformations. First, the raw summary statistics are brought closer to normality with one parameter (without shift) Box-Cox transformation applied independently to each statistic. Many of the proposed summary statistics are zero-inflated, which would make estimation of the parameter $\lambda$ of the Box-Cox transformation unreliable. Because of this, fixed value $\lambda = 0.5$ is used. After the Box-Cox transformation, the summary statistics are centered and standardized to have unit variance. Second, the summary statistics are transformed with partial least squares (PLS) to produce orthogonal summary statistics while maximizing covariance with the parameters (*Wegmann et al., 2009*). The PLS is applied to all of the simulated parameter values and datasets. If applied to a set of summary statistics of dimension $k$, the maximum number of transformed statistics is *k-1*. In the analyses we use dimensionality of 20, which is larger than the number of parameters, but still substantially lower than the number of original summary statistics. Finally, local linear regression adjustment is utilized for the parameters in the set of accepted values (*Beaumont et al., 2002*). The parameter values are transformed based on the distances of the summary statistics to the summaries of the observed data.

## Summary statistics

We use the following raw summary statistics (36 in total):

- Number of patches with infected plants.

- Number of patches with coinfected plants.
- Mean and standard deviation of pathogen population size categories.
- Parameter estimates and standard deviation of residuals from linear regression of covariates against the pathogen population size category.
- Mean and standard deviation of the disease prevalence categories.
- Parameter estimates and standard deviation of residuals from linear regression of covariates against the disease prevalence category.
- Parameter estimates and standard deviation of residuals from binomial regression of covariates against the number of infected plants in a patch.
- Parameter estimates and standard deviation of residuals from binomial regression of covariates against the number of coinfected plants in a patch.
- Parameter estimates and standard deviation of residuals from Poisson regression of covariates against the number of observed MLGs in a patch.

## Effect of the modeling assumptions

To assess the robustness of the results, we tested the effect that the modeling assumptions have on the posterior distributions of the parameters. Studying all aspects of the model is obviously not possible, but we chose two assumptions, namely the order $\sigma_{i,j}$ in which the mildew strains try to infect the plant, and the maximum number of strains in a patch $M_{max}$.

We used three different distributions for the order $\sigma_{i,j}$:

1. Uniform distribution among all permutations of $M_i$ items.
2. Fixed order based on decreasing $\mu_{i,k}$s, i.e. MLGs with highest prevalences being first to infect.
3. Simulate first $\delta_{i,j} = (\delta_{i,j,1}, \ldots, \delta_{i,j,M_i})$ from Dirichlet distribution with parameters $\kappa \text{logit}^{-1}(\mu_{i,k})$ and choose the order based on decreasing values of $\delta_{i,j,k}$.

The last distribution is intermediate between the first two, and the value of $\kappa$ dictates how much the prevalences of individual MLGs affect the order. If $\kappa$ is close to 0 then the distribution is close to uniform, and if $\kappa$ goes to infinity the order is the same as second choice above. We use a value $\kappa = 5$ in the analyses.

We analyzed the year 2012 data four times with different assumptions. For the first three we used $M_{max} = 30$ and each of the distributions shown above for $\sigma_{i,j}$. For the last one we set $M_{max} = 100$ and the uniform distribution for $\sigma_{i,j}$. In each analysis we performed 50,000 simulations and used 500 closest for the posterior.

## Accuracy of inference

As a measure of goodness of fit the root mean squared error (RMSE) was computed independently for each parameter. RMSE for a parameter $\theta$ is computed as

$$RMSE(\theta|A) = \sqrt{\frac{1}{N} \sum_{j \in A} (\theta_j - \theta_t)^2},$$

## Data availability and repeatability of analysis

All data and scripts used to perform the analyses presented in this paper are available in the git repository at https://github.com/ComputerBlue/FungalSex (*Laine, 2019*; copy archived at https://github.com/elifesciences-publications/FungalSex).

## Acknowledgements

We would like to acknowledge the numerous students who have been involved in the survey and sampling of powdery mildew populations in the Åland Islands in 2012–15. K Raveala supervised the field surveys. B McDonald and T Giraud provided comments on an earlier version of the manuscript. Comments from K King, D Weigel, E Decaestecker and an anonymous reviewer during the review process helped significantly improve the manuscript. The DNA extractions and SNP genotyping were carried out at Institute of Biotechnology and Finnish Institute of Molecular Medicine at University of Helsinki, respectively. This work was funded by grants from the Academy of Finland (296686),

Jane and Aatos Erkko Foundation, and the European Research Council (Starting Grant PATHEVOL 281517 and Consolidator Grant RESISTANCE 724508) to A-LL.

## Additional information

### Funding

| Funder | Grant reference number | Author |
|---|---|---|
| European Research Council | 281517 | Anna-Liisa Laine |
| European Research Council | 724508 | Anna-Liisa Laine |
| Academy of Finland | 296686 | Anna-Liisa Laine |

The funders had no role in study design, data collection and interpretation, or the decision to submit the work for publication.

### Author contributions

Anna-Liisa Laine, Conceptualization, Resources, Supervision, Funding acquisition, Writing—original draft; Benoit Barrès, Data curation, Methodology, BB oversaw the large-scale genotyping study; Elina Numminen, Data curation, Formal analysis, Visualization, Methodology, Writing—original draft, EN carried out the spatial statistical analyses; Jukka P Siren, Software, Validation, Visualization, Methodology, Writing—original draft, JS implemented the Approximate Bayesian Computation model

### Author ORCIDs

Anna-Liisa Laine (iD) https://orcid.org/0000-0002-0703-5850
Benoit Barrès (iD) https://orcid.org/0000-0002-6777-0275

### Decision letter and Author response

Decision letter https://doi.org/10.7554/eLife.47091.020
Author response https://doi.org/10.7554/eLife.47091.021

## Additional files

### Supplementary files

• Supplementary file 1. Characteristics of the pathogen metapopulation for each year. MLG: MultiLocus Genotype.
DOI: https://doi.org/10.7554/eLife.47091.017

• Transparent reporting form
DOI: https://doi.org/10.7554/eLife.47091.018

### Data availability

All data and scripts used to perform the analyses presented in this paper are available in the git repository at https://github.com/ComputerBlue/FungalSex (copy archived at https://github.com/elifesciences-publications/FungalSex).

The following datasets were generated:

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
