## [Decision Letter]

Thank you for submitting your article "A direct ecological benefit of outcrossing in an obligate pathogen" for consideration by *eLife*. Your article has been reviewed by three peer reviewers, including Kayla King as guest Reviewing Editor, and the evaluation has been overseen by Detlef Weigel as the Senior Editor. The following individual involved in review of your submission has agreed to reveal their identity: Ellen Decaestecker (Reviewer #3).

The reviewers have discussed the reviews with one another and the Reviewing Editor has drafted this decision to help you prepare a revised submission.

This work examines the advantages to outcrossing via coinfection in a plant pathogen. It comprises a thorough and complete analysis of the opportunities for outcrossing due to coinfection using an extensive, multi-year field data set from 4000 plant host populations. This study advances our understanding of the benefits of sex in organisms with the most common life-style in nature, and across heterogeneous landscapes (and environmental conditions).

Summary:

The advantages of outcrossing in nature remain one of the biggest questions tackled by evolutionary biologists. Laine and colleagues address this question, in terms of overwintering success, using a phytopathogen which maintains mixed reproductive strategies (outcrossing and asexual reproduction/selfing) on its plant host. They use the powdery mildew fungus *Podosphaera plantaginis*, which is an obligate pathogen of *Plantago lanceolate*, to first test how coinfection varies across pathogen populations (coinfection is needed for outcrossing), they then examine whether novel pathogen multilocus genotypes are associated with coinfection (which would indicate outcrossing), and finally whether coinfection affects overwintering success. Using an impressively-sized and natural field study analysing almost 3000 pathogen populations over four sampling years, the authors use Spatial Bayesian models to assess disease dynamics and genotypic diversity data. They find that coinfection is common and spatially variable, that where there are high levels of coinfection there are more novel pathogen MLGs, and that there is a positive effect of coinfection on overwintering success. The authors provide evidence that a pathogen can ecologically benefit from outcrossing, and this result opens the door to more detailed experimental understanding of this phenomenon in the future.

Overall, this manuscript is interesting and timely, and contributes substantially to the debate on the evolution of sex. We have the following suggestions to improve the manuscript.

Essential revisions:

The Introduction would benefit from hypotheses and a mention on why the benefits of sex need exploration in pathogens/parasites. This study provides a very interesting and novel approach to questions surrounding the evolution of sex. What makes parasites similar/different from the free-living species in which the benefits of sex have already been examined? It may be valuable to integrate literature on non-parasite sexual recombination versus selfing and the fitness advantage of that in non-parasite individuals, e.g. in cyclical parthenogenetic invertebrates with resting stages.

Discussion of the results in the context of infections and evolution of sex would bolster the generality and significance of the study. At the moment, the authors undersell their findings in the final paragraph of the Discussion and fail to put them in a bigger context for infectious disease, epidemics, and evolution of sex.

---

## [Author Response]

Essential revisions:The Introduction would benefit from hypotheses and a mention on why the benefits of sex need exploration in pathogens/parasites. This study provides a very interesting and novel approach to questions surrounding the evolution of sex. What makes parasites similar/different from the free-living species in which the benefits of sex have already been examined? It may be valuable to integrate literature on non-parasite sexual recombination versus selfing and the fitness advantage of that in non-parasite individuals, e.g. in cyclical parthenogenetic invertebrates with resting stages.

We thank the editors for this suggestions, and we have now completely re-written the first two paragraphs to provide a more concise theoretical framework for our study. We first outline the predictions and empirical support for the Red Queen Hypothesis in explaining the maintenance of sex in free-living hosts. We then clarify that coevolution should in fact be even more critical for maintaining outcrossing in parasite populations that need to adapt to their ever-changing host populations. We now cover the limited theoretical literature on the role of coevolution in maintaining sex in parasite populations (Introduction).

We also consider in more depth the fitness advantages of asexual vs. sexual strategies especially in facultatively sexual organisms where sexual offspring provide ecological functions such as dormancy and dispersal. We relate this to homothallic species where the same functions may be achieved via selfing or outcrossing but still outcrossing is frequently found (Introduction).

Discussion of the results in the context of infections and evolution of sex would bolster the generality and significance of the study. At the moment, the authors undersell their findings in the final paragraph of the Discussion and fail to put them in a bigger context for infectious disease, epidemics, and evolution of sex.

We thank to the editors for this suggestion and agree that we had not provided an in depth consideration of the broader relevance of our findings. We’ve now added a new paragraph in the Discussion where we consider both the evolutionary and epidemiological implications of our findings for pathogens more generally.